# PREVENTING VALUE FUNCTION COLLAPSE IN ENSEMBLE Q-LEARNING BY MAXIMIZING REPRESENTATION DIVERSITY

## ABSTRACT

The first deep RL algorithm, DQN, was limited by the overestimation bias of the learned Q-function. Subsequent algorithms proposed techniques to reduce this problem, without fully eliminating it. Recently, the Maxmin and Ensemble Q-learning algorithms used the different estimates provided by ensembles of learners to reduce the bias. Unfortunately, these learners can converge to the same point in the parametric or representation space, falling back to the classic single neural network DQN. In this paper, we describe a regularization technique to maximize diversity in the representation space in these algorithms. We propose and compare five regularization functions inspired from economics theory and consensus optimization. We show that the resulting approach significantly outperforms the Maxmin and Ensemble Q-learning algorithms as well as non-ensemble baselines.

## 1 INTRODUCTION

Q-learning (Watkins, 1989) and its deep learning based successors inaugurated by DQN (Mnih et al., 2015) are model-free, value function based reinforcement learning algorithms. Their popularity stems from their intuitive, easy-to-implement update rule derived from the Bellman equation. At each time step, the agent updates its Q-value towards the expectation of the current reward plus the value corresponding to the maximal action in the next state. This state-action value represents the maximum sum of reward the agent believes it could obtain from the current state by taking the current action. Unfortunately (Thrun & Schwartz, 1993; van Hasselt, 2010) have shown that this simple rule suffers from *overestimation bias*: due to the maximization operator in the update rule, positive and negative errors do not cancel each other out, but positive errors accumulate. The overestimation bias is particularly problematic under function approximation and have contributed towards learning sub-optimal policies (Thrun & Schwartz, 1993; Szita & Lőrincz, 2008; Strehl et al., 2009).

A possible solution is to introduce *underestimation* bias in the estimation of the Q-value. Double Q-learning (van Hasselt, 2010) maintains two independent state-action value estimators (Q-functions). The state-action value of estimator one is calculated by adding observed reward and maximal state-action value from the other estimator. Double DQN (Hado van Hasselt et al., 2016) applied this idea using neural networks, and was shown to provide better performance than DQN. More recent actor-critic type deep RL algorithms such as TD3 (Fujimoto et al., 2018) and SAC (Haarnoja et al., 2018) also use two Q function estimators (in combination with other techniques).

Other approaches such as EnsembleDQN (Anschel et al., 2017) and MaxminDQN (Lan et al., 2020) maintain *ensembles* of Q-functions to estimate an unbiased Q-function. EnsembleDQN estimates the state-action values by adding the current observed reward and the maximal state-action value from the average of Q-functions from the ensemble. MaxminDQN creates a proxy Q-function by selecting the minimum Q-value for each action from all the Q-functions and using the maximal state-action value from the proxy Q-function to estimate an unbiased Q-function. Both EnsembleDQN and MaxminDQN have been shown to perform better than Double DQN. The primary insight of this paper is that the performance of ensemble based methods is *contingent on maintaining sufficient diversity in the representation space* between the Q-functions in the ensembles. If the Q-functions in the ensembles converge to a common representation (we will show that this is the case in many scenarios), the performance of these approaches significantly degrades.

In this paper we propose to use cross-learner regularizers to prevent the collapse of the representation space in ensemble-based Q-learning methods. Intuitively, these representations capture an inductive bias towards more diverse representations. We have investigated five different regularizers. The mathematical formulation of four of the regularizers correspond to inequality measures borrowed from economics theory. While in economics, high inequality is seen as a negative, in this case we use the metrics to encourage inequality between the representations. The fifth regularizer is inspired from consensus optimization.

There is a separate line of reinforcement learning literature where ensembles are used to address several different issues (Chen et al., 2017; Chua et al., 2018; Kurutach et al., 2018; Lee et al., 2020; Osband et al., 2016) such as exploration and error propagation but we limit our solution to algorithms addressing the overestimation bias problem only.

To summarize, our contributions are following:

1. We show that high representation similarity between neural network based Q-functions leads to decline in performance in ensemble based Q-learning methods.

2. To mitigate this, we propose five regularizers based on inequality measures from economics theory and consensus optimization that maximize representation diversity between Q-functions in ensemble based Q-learning methods.

3. We show that applying the proposed regularizers to the MaxminDQN and EnsembleDQN methods can lead to significant improvement in performance over a variety of benchmarks.

## 2 BACKGROUND

Reinforcement learning considers an agent as a Markov Decision Process (MDP) defined as a five element tuple $(\mathcal{S}, \mathcal{A}, P, r, \gamma)$, where $\mathcal{S}$ is the state space, $\mathcal{A}$ is the action space, $P : \mathcal{S} \times \mathcal{A} \times \mathcal{S} \to [0, 1]$ are the state-action transition probabilities, $r : \mathcal{S} \times \mathcal{A} \times \mathcal{S} \to \mathbb{R}$ is the reward mapping and $\gamma \to [0, 1]$ is the discount factor. At each time step $t$ the agent observes the state of the environment $s_t \in \mathcal{S}$ and selects an action $a_t \in \mathcal{A}$. The effect of the action triggers a transition to a new state $s_{t+1} \in \mathcal{S}$ according to the transition probabilities $P$, while the agent receives a scalar reward $R_t = r(s_t, a_t, s_{t+1})$. The goal of the agent is to learn a policy $\pi$ that maximizes the expectation of the discounted sum of future rewards.

One way to implicitly learn the policy $\pi$ is the Q-learning algorithm that estimates the expected sum of rewards of state $s_t$ if we take the action $a_t$ by solving the Bellman equation

$$Q^* (s_t, a_t) = \mathbb{E} \left[ R_t + \max_{a' \in \mathcal{A}} Q^* (s_{t+1}, a') \right]$$

The implicit policy $\pi$ can extracted by acting greedily with respect to the optimal Q-function: $\arg \max_{a \in \mathcal{A}} Q^* (s, a)$. One possible way to estimate the optimal Q-value is by iteratively updating it for sampled states $s_t$ and action $a_t$ using

$$Q^* (s_t, a_t) \leftarrow Q^* (s_t, a_t) + \alpha (Y_t - Q^* (s_t, a_t)) \quad \text{where } Y_t = R_t + \max_{a' \in \mathcal{A}} Q^* (s_{t+1}, a')$$

where $\alpha$ is the step size and $Y_t$ is called the target value. While this algorithm had been initially studied in the context of a tabular representation of Q for discrete states and actions, in many practical applications the Q value is approximated by a learned function. Since the emergence of deep learning, the preferred approximation technique is based on a deep neural network. DQN (Mnih et al., 2015), had demonstrated super-human performance in Atari Games, but required a very large number of training iterations. From this baseline, subsequent algorithms improved both the learning speed and achievable performance, with one of the main means for this being techniques to reduce the overestimation bias of the Q-function.

EnsembleDQN (Anschel et al., 2017) uses an ensemble of $N$ neural networks to estimate state-action values and uses their average to reduce both overestimation bias and estimation variance. Formally,

the target value for EnsembleDQN is calculated using

$$Q_E\left(\cdot\right) = \frac{1}{N}\sum_{i=1}^{N}Q_i\left(\cdot\right)$$

$$Y_t^E = R_t + \max_{a'\in\mathcal{A}}Q_E\left(s_{t+1}, a'\right) \tag{1}$$

More recent, MaxminDQN (Lan et al., 2020) addresses the overestimation bias using order statistics, using the ensemble size $N$ as a hyperparameter to tune between underestimating and overestimating bias. The target value for MaxminDQN is calculated using

$$Q_M\left(\cdot, \cdot\right) = \min_{i=1,\ldots,N}Q_i\left(\cdot, \cdot\right)$$

$$Y_t^M = R_t + \max_{a'\in\mathcal{A}}Q_M\left(s_{t+1}, a'\right) \tag{2}$$

## 3 RELATED WORK

**Techniques to Address Overestimation Bias in RL:**  Addressing overestimation bias is a long standing research topic not only in reinforcement learning but other fields of science such as economics and statistics. It is commonly known as *max-operator bias* in statistics (D'Eramo et al., 2017) and as *the winner's curse* in economics (Thaler, 2012; Smith & Winkler, 2006). To address this, (van Hasselt, 2010) proposed Double Q-learning, subsequently adapted to a neural network based function approximators as Double DQN (Hado van Hasselt et al., 2016). Alternatively, (Zhang et al., 2017; Lv et al., 2019) proposed weighted estimators of Double Q-learning and (Lee et al., 2013) introduced a bias correction term. Other approaches to address the overestimation are based on averaging and ensembling. Techniques include averaging Q-values from previous $N$ versions of the Q-network (Anschel et al., 2017), taking linear combinations of $\min$ and $\max$ over the pool of Q-values (Kumar et al., 2019), or using a random mixture from the pool (Agarwal et al., 2019).

**Regularization in Reinforcement Learning:**  Regularization in reinforcement learning has been used to perform effective exploration and learning generalized policies. For instance, (Grau-Moya et al., 2019) uses mutual-information regularization to optimize a prior action distribution for better performance and exploration, (Cheng et al., 2019) regularizes the policy $\pi(a|s)$ using a control prior, (Galashov et al., 2019) uses temporal difference error regularization to reduce variance in Generalized Advantage Estimation (Schulman et al., 2016). Generalization in reinforcement learning refers to the performance of the policy on different environment compared to the training environment. For example, (Farebrother et al., 2018) studied the effect of $L^2$ norm on DQN on generalization, (Tobin et al., 2017) studied generalization between simulations vs. the real world, (Pattanaik et al., 2018) studied parameter variations and (Zhang et al., 2018) studied the effect of different random seeds in environment generation.

**Representation Similarity:**  Measuring similarity between the representations learned by different neural networks is an active area of research. For instance, (Raghu et al., 2017) used Canonical Correlation Analysis (CCA) to measure the representation similarity. CCA find two basis matrices such that when original matrices are projected on these bases, the correlation is maximized. (Raghu et al., 2017; Mroueh et al., 2015) used truncated singular value decomposition on the activations to make it robust for perturbations. Other work such as (Li et al., 2015) and (Wang et al., 2018) studied the correlation between the neurons in the neural networks.

## 4 MAXIMIZING REPRESENTATION DIVERSITY IN ENSEMBLE-BASED DEEP Q-LEARNING

The work described in this paper is based on the conjecture that while ensemble-based deep Q-learning approaches aim to reduce the overestimation bias, this only works to the degree that the neural networks in the ensemble use diverse representations. If during training, these networks collapse to closely related representations, the learning performance decreases. From this idea, we propose to use regularization techniques to maximize representation diversity between the networks of the ensemble.

## 4.1 REPRESENTATION SIMILARITY MEASURE

Let $X \in \mathbb{R}^{n \times p_1}$ denote a matrix of activations of $p_1$ neurons for $n$ examples and $Y \in \mathbb{R}^{n \times p_2}$ denote a matrix of activations of $p_2$ neurons for the same $n$ examples. Furthermore, we consider $K_{ij} = k(x_i, x_j)$ and $L_{ij} = l(y_i, y_j)$ where $k$ and $l$ are two kernels.

Centered Kernel Alignment (CKA) (Kornblith et al., 2019; Cortes et al., 2012; Cristianini et al., 2002) is a method for comparing representations of neural networks, and identifying correspondences between layers, not only in the same network but also on different neural network architectures. CKA is a normalized form of Hilbert-Schmidt Independence Criterion (HSIC) (Gretton et al., 2005). Formally, CKA is defined as:

$$\text{CKA}(K, L) = \frac{\text{HSIC}(K, L)}{\sqrt{\text{HSIC}(K, K) \cdot \text{HSIC}(L, L)}}$$

HSIC is a test statistic for determining whether two sets of variables are independent. The empirical estimator of HSIC is defined as:

$$\text{HSIC}(K, L) = \frac{1}{(n-1)^2} \text{tr}(KHLH)$$

where $H$ is the centering matrix $H_n = I_n - \frac{1}{n} \mathbf{1} \mathbf{1}^T$.

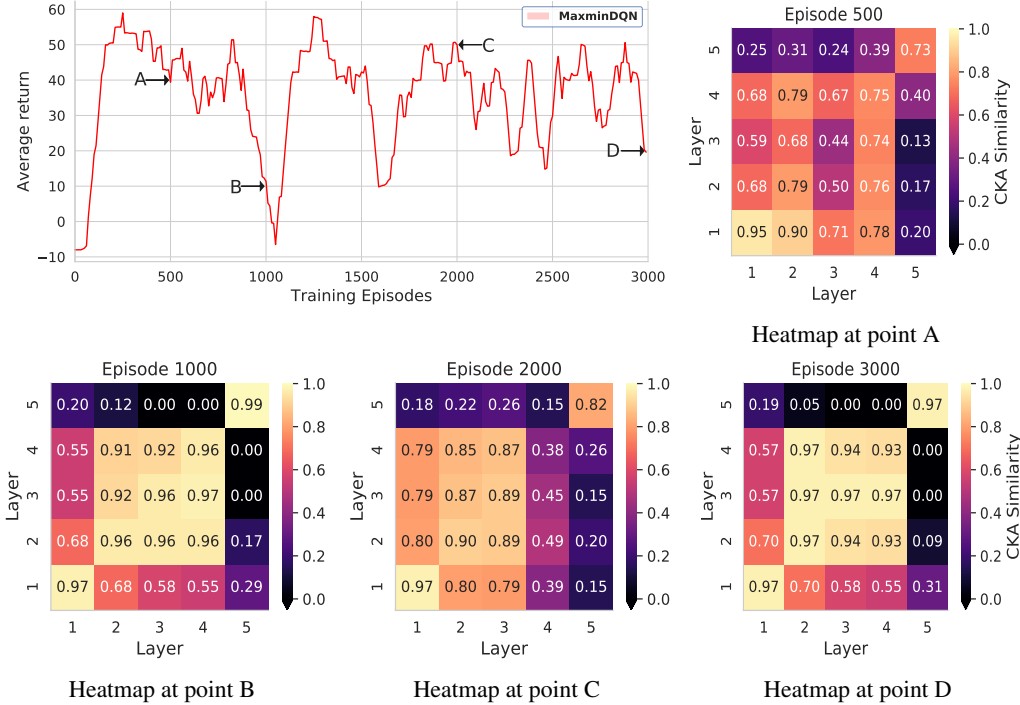

Figure 1: The training graph and CKA similarity heatmaps of a MaxminDQN agent with 2 neural networks. The letters on the plot show the time when CKA similarities were calculated. Heatmaps at A and C have relatively low CKA similarity and have relatively higher average return as compared to heatmaps at point B and D that have extremely high similarity across all the layers.

## 4.2 CORRELATION BETWEEN PERFORMANCE AND REPRESENTATION SIMILARITY

The work in this paper starts from the conjecture that high representation similarity between neural networks in an ensemble-based Q-learning technique correlates to poor performance. To empirically verify our hypothesis, we trained a MaxminDQN agent with two neural networks on the Catcher

environment (Qingfeng, 2019) for about 3000 episodes ($5 \times 10^6$ training steps) and calculated the CKA similarity with a linear kernel after every 500 episodes. The training graph along with the CKA similarity heatmaps are shown in Figure 1. Notably at episode 500 (heatmap A) and episode 2000 (heatmap C), the representation similarity between neural networks is low but the average return is relatively high. In contrast, at episode 1000 (heatmap B) and episode 3000 (heatmap D) the representation similarity is highest but the average return is lowest.

Additionally, in Appendix A.1, we performed a regression experiment to demonstrate that when two neural networks trained on same data, despite having different architecture, learning rate and batch size can learn almost identical representations. This experiment also demonstrates that random initialization of neural networks enforces diversity is a *misconception*.

### 4.3 REGULARIZATION FOR MAXIMIZING REPRESENTATION DIVERSITY

In order to maximize the representation diversity, we propose to regularize the training algorithm with an additional criteria that favors diversity in the representation space. In the following, $N$ is the number of neural networks in the ensemble, $\ell_i$ is the $L^2$ norm of the $i$-th neural network's parameters, $\bar{\ell}$ is the mean of all the $L^2$ norms and $\ell$ is the list of all the $L^2$ norms.

The first four metrics we consider are based on inequality measures from economic theory. While in economics, inequality is usually considered something to be avoided, in our case we aim to increase inequality (and thus, representation diversity).

The **Atkinson Index** (Atkinson et al., 1970) measures income inequality and is useful in identifying the end of the distribution that contributes the most towards the observed inequality. Formally, it is defined as

$$A_\epsilon = \begin{cases} 1 - \frac{1}{\bar{\ell}}\left(\frac{1}{N}\sum_{i=1}^{N}\ell_i^{1-\epsilon}\right)^{\frac{1}{1-\epsilon_{at}}}, & \text{for } 0 \leq \epsilon_{at} \neq 1, \\ 1 - \frac{1}{\bar{\ell}}\left(\frac{1}{N}\prod_{i=1}^{N}\ell_i\right)^{\frac{1}{N}}, & \text{for } \epsilon_{at} = 1, \end{cases} \tag{3}$$

where $\epsilon_{at}$ is the inequality aversion parameter used to tune the sensitivity of the measured change. When $\epsilon_{at} = 0$, the index is more sensitive to the changes at the upper end of the distribution, while the index becomes more sensitive towards the change at the lower end of the distribution when $\epsilon_{at}$ approaches 1.

The **Gini coefficient** (Allison, 1978) is a statistical measure of the wealth distribution or income inequality among a population and defined as the half of the relative mean absolute difference:

$$G = \frac{\sum_{i=1}^{N}\sum_{j=1}^{N}|\ell_i - \ell_j|}{2N^2\bar{\ell}} \tag{4}$$

The Gini coefficient is more sensitive to deviation around the middle of the distribution than at the upper or lower part of the distribution.

The **Theil index** (Johnston, 1969) measures redundancy, lack of diversity, isolation, segregation and income inequality among a population. Using the Theil index is identical to measuring the redundancy in information theory, defined as the maximum possible entropy of the data minus the observed entropy:

$$T_T = \frac{1}{N}\sum_{i=1}^{N}\frac{\ell_i}{\bar{\ell}}\ln\frac{\ell_i}{\bar{\ell}} \tag{5}$$

The **variance of logarithms** (Ok & Foster, 1997) is a widely used measure of dispersion with natural links to wage distribution models. Formally, it is defined as:

$$V_L(\ell) = \frac{1}{N}\sum_{i=1}^{N}[\ln\ell_i - \ln g(\ell)]^2 \tag{6}$$

where g($\ell$) is the geometric mean of $\ell$ defined as $(\prod_{i=1}^{N}\ell_i)^{1/N}$.

The final regularization method we use is inspired from consensus optimization. In a consensus method (Boyd et al., 2011), a number of models are independently optimized with their own task-specific parameters, and the tasks communicate via a penalty that encourages all the individual solutions to converge around a common value. Formally, it is defined as

$$M = \|\bar{\ell} - \ell_i\|^2 \tag{7}$$

We will refer this regularizer as MeanVector throughout this paper.

### 4.4 Training Algorithm

Using the regularization functions defined above, we can develop diversity-regularized variants of the the MaxminDQN and EnsembleDQN algorithms. The training technique is identical to the algorithms described in (Lan et al., 2020) and (Anschel et al., 2017), with a regularization term added to the loss of the Q-functions. The loss term for $i$-th Q-function with parameters $\psi_i$ is:

$$\mathcal{L}(\psi_i) = \mathbb{E}_{s,a,r,s'}\left[\left(Q^i_\psi(s,a) - Y\right)^2\right] - \lambda \mathcal{I}(\ell_i, \boldsymbol{\ell}),$$

where $Y$ is the target value calculated using either Equation (1) or Equation (2) depending on the algorithm, $\mathcal{I}$ is the regularizer of choice from the list above and $\lambda$ is the regularization weight. Notice that the regularization term appears with a negative sign, as the regularizers are essentially inequality metrics that we want to maximize. For completeness, the algorithm are shown in Appendix B.

## 5 Experiments

### 5.1 Training Curves

We chose three environments from PyGames(Qingfeng, 2019) and MinAtar (Young & Tian, 2019): Catcher, Pixelcopter and Asterix. These environments were used by the authors in the MaxminDQN (Lan et al., 2020) paper. We reused all the hyperparameter settings from (Lan et al., 2020) except the number of neural networks, which we limited to four and trained each solution for five fixed seeds. For the regularization weight $\lambda$, we chose the best value from $\{10^{-5}, 10^{-6}, 10^{-7}, 10^{-8}\}$. *The baselines were also fine-tuned.* The complete list of training parameters can be found in Appendix E.

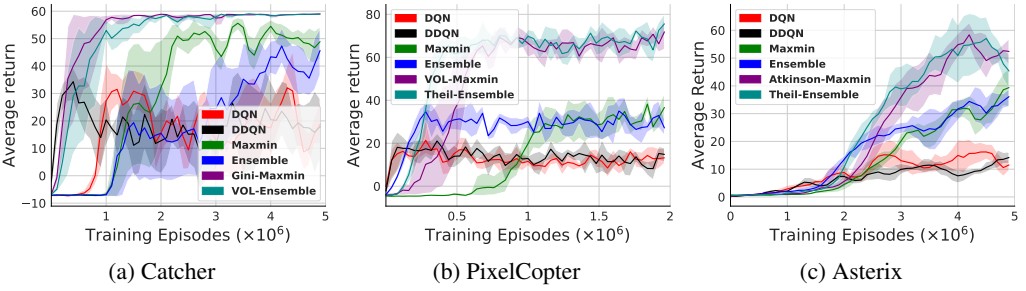

|  (a) Catcher | (b) PixelCopter | (c) Asterix |

Figure 2: Training curves and 95% confidence interval (shaded area) for the best augmented variants for MaxminDQN and EnsembleDQN together with baseline algorithms.

Figure 2 shows the training curves for the three environments. To avoid crowding the figures, for each environment and baseline algorithm (MaxminDQN and EnsembleDQN) we only plotted the regularized version which performed the best. We also show as baseline the original MaxminDQN and EnsembleDQN, as well as the DQN and DDQN algorithms. For the Catcher environment, both Gini-MaxminDQN and VOL-EnsembleDQN were able to quickly reach the optimal performance and stabilized after $2 \times 10^6$ training steps while the baseline MaxminDQN reached its maximum performance after $3.5 \times 10^6$ training steps but went down afterwards. Similarly, the baseline EnsembleDQN reached its maximum performance after $4 \times 10^6$ training steps, with the performance fluctuating with continued training. For the PixelCopter environment, VOL-MaxminDQN and Theil-EnsembleDQN were slower in the initial part of the learning that some of the other approaches, but

over time they achieved at least double return compared to the other approaches. Similarly, for the Asterix environment, Atkinson-MaxminDQN and Theil-EnsembleDQN lagged in training for about $1 \times 10^6$ training steps but after that they achieved at least $50\%$ higher return compared to the baselines. Full results together with CKA similarity heatmaps are shown in Appendix C.

## 5.2 T-SNE VISUALIZATIONS

To visualize the impact of the regularization, Figure 3 shows t-SNE (van der Maaten & Hinton, 2008) visualization of the activations of the last layer of the trained networks. Figure 3a show the network trained for the Catcher environment, while Figure 3b, the network trained for the PixelCopter environment. The upper row of the figure shows the original, unregularized models, while the lower row a regularized version. For all combinations, we find that the activations from the original MaxminDQN and EnsembleDQN versions do not show any obvious pattern, while the regularized ones show distinct clusters. An additional benefit of t-SNE visualizations over CKA similarity heatmaps is that the CKA similarity heatmaps are useful to show representation similarity between two neural networks, but they become counter intuitive as the number of neural networks increases. More t-SNE visualizations for four neural network experiments are shown in Appendix C.3.

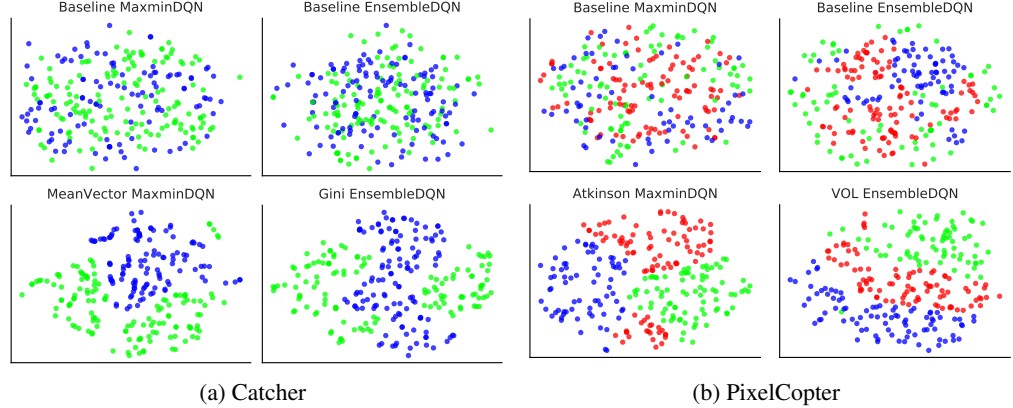

(a) Catcher                                              (b) PixelCopter

Figure 3: Clustering last layer activations from Catcher and PixelCopter after processing them with t-SNE to map them in 2D. The regularized variants have visible clusters while the baseline MaxminDQN and EnsembleDQN activations are mixed together with no visible pattern.

## 5.3 STATISTICAL ANALYSIS

**What is the impact of the regularization on the performance?** Similar to the approach taken by (Liu et al., 2020), we performed a $z$-score test to rigorously evaluate the improvement of regularization over baseline solutions. The $z$-score is also known as "standard score", the signed fractional number of standard deviations by which the value of a data point is above the mean value. A regularizer's $z$-score roughly measures its relative performance among others. For each algorithm, environment and neural network setting, we calculated the $z$-score for each regularization method and the baseline by treating results all the results as a populations. For example, to find out which EnsembleDQN with two neural networks is best for the Catcher environment, we took the average reward of 10 episodes for each experiment ($(5+1) \times 5$ seeds) and treated it as a population. Finally, we averaged the $z$-scores to generate the final result presented in Table 1. In terms of improved performance, all the regularizers have achieved significant improvement over the baselines for all three environments. The $z$-scores for four neural network experiments are shown in Appendix C.4.

**Is the improvement statistically significant?** We collected the $z$-scores from the previous section and performed the Welch's $t$-test with the corresponding $z$-scores produced by the baseline. The resulting $p$-values are presented in Table 2. From the results, we observed that the improvement introduced from regularization is statistically significant ($p < 0.05$) in almost all the cases.

Table 1: Averaged $z$-scores for each regularization method.

| Reg | Ensemble N=2 | | | Maxmin N=2 | | | Ensemble N=3 | | | Maxmin N=3 | | |
|---|---|---|---|---|---|---|---|---|---|---|---|---|
| | Catcher | Copter | Asterix | Catcher | Copter | Asterix | Catcher | Copter | Asterix | Catcher | Copter | Asterix |
| Baseline | -2.125 | -2.044 | -1.868 | -2.143 | -2.031 | -1.730 | -1.931 | -2.042 | -1.634 | -1.702 | -2.115 | -1.478 |
| Atkinson | 0.419 | 0.450 | 0.187 | 0.353 | 0.402 | **0.459** | 0.379 | 0.292 | 0.197 | 0.344 | 0.380 | **0.795** |
| Gini | 0.422 | **0.202** | 0.659 | **0.539** | 0.446 | 0.201 | 0.349 | 0.231 | **0.449** | 0.319 | **0.624** | 0.068 |
| MeanVector | 0.426 | 0.359 | 0.372 | 0.529 | 0.079 | 0.315 | **0.403** | **0.763** | 0.227 | **0.347** | 0.316 | 0.174 |
| Theil | 0.425 | 0.358 | **0.767** | 0.198 | **0.576** | 0.364 | 0.402 | 0.284 | 0.429 | 0.341 | 0.499 | 0.068 |
| VOL | **0.433** | 0.315 | 0.341 | 0.522 | 0.526 | 0.392 | 0.397 | 0.471 | 0.149 | 0.332 | 0.297 | 0.372 |

Table 2: $P$-values from Welch's $t$-test comparing the $z$-scores of regularization and baseline

| Reg | Ensemble N=2 | | | Maxmin N=2 | | | Ensemble N=3 | | | Maxmin N=3 | | |
|---|---|---|---|---|---|---|---|---|---|---|---|---|
| | Catcher | Copter | Asterix | Catcher | Copter | Asterix | Catcher | Copter | Asterix | Catcher | Copter | Asterix |
| Atkinson | 0.002 | 0.000 | 0.003 | 0.000 | 0.000 | 0.000 | 0.019 | 0.000 | 0.000 | 0.061 | 0.000 | 0.003 |
| Gini | 0.002 | 0.000 | 0.001 | 0.000 | 0.000 | 0.000 | 0.002 | 0.003 | 0.003 | 0.061 | 0.000 | 0.013 |
| MeanVector | 0.002 | 0.000 | 0.000 | 0.000 | 0.008 | 0.016 | 0.019 | 0.000 | 0.000 | 0.061 | 0.000 | 0.030 |
| Theil | 0.002 | 0.002 | 0.002 | 0.000 | 0.000 | 0.002 | 0.019 | 0.001 | 0.006 | 0.061 | 0.001 | 0.026 |
| VOL | 0.002 | 0.000 | 0.000 | 0.000 | 0.005 | 0.001 | 0.019 | 0.000 | 0.017 | 0.060 | 0.000 | 0.008 |

## 6  IDENTICAL LAYERS EXPERIMENT

To test the limits of the regularizers, we initialized, each layer of each neural network with the same fixed seed. This initialization enforces maximum representation similarity and is considered the worst case scenario for ensemble based learning methods. We performed this experiment on all three environments and used the same seeds and hyperparameters that were used for the main experiments. The training curves are shown in Figure 4. Notably, the results from the baseline MaxminDQN and EnsembleDQN on both Catcher and PixelCopter environments are similar to the main results. For the Catcher environment, both Gini-MaxminDQN and Theil-EnsembleDQN were slow in learning for about $2 \times 10^6$ training steps but both solutions were able to achieve the optimal performance by the end of training. Similarly for PixelCopter environment, the VOL-MaxminDQN was slow in learning till $1.5 \times 10^6$ training steps but it was able to outperform the baseline results and achieved optimal performance. The complete training plots for these experiments are shown in Appendix D.

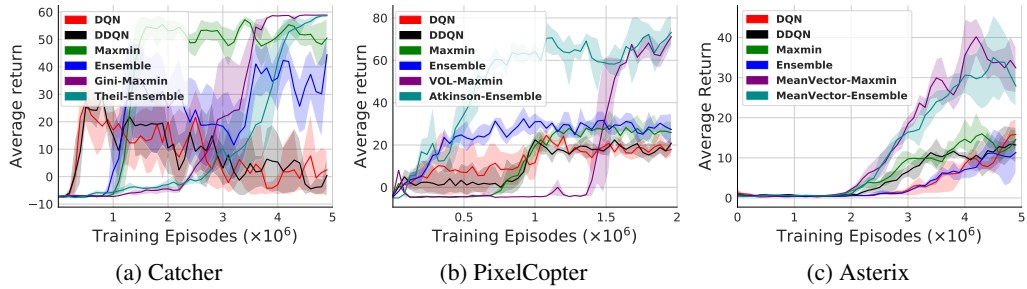

(a) Catcher                          (b) PixelCopter                          (c) Asterix

Figure 4: Training plots representing the best results from each solution for Catcher, PixelCopter and Asterix environment when the layers of the neural networks were initialized with one fixed seed.

## 7  CONCLUSION

In this paper we showed that high representation similarity between the Q-functions in ensemble based Q-learning algorithms such as MaxminDQN and EnsembleDQN leads to a decline in learning performance. To mitigate this, we proposed a regularization approach using five different metrics to maximize the diversity in the representation space of the Q-functions. Experiments have shown that our solution outperforms baseline MaxminDQN and EnsembleDQN in standard training settings as well as in scenarios where the parameters of the neural layers were initialized using one fixed seed.

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

# A Supplementary Material

## A.1 Motivating Example to Demonstrate Similarity Between Neural Networks

We performed a regression experiment in which we learnt a sine wave function using two different three layered fully connected neural networks with 64 and 32 neurons in each hidden layer with ReLU. The neural networks were initialized using different seeds and were trained using different batch sizes $(512, 128)$ and learning rates $(1e-4, 1e-3)$. The Figure 5a shows the learnt functions while Figure 5b represents their CKA similarity heatmap before and after training. The odd numbered layers represent pre-ReLU activations while the even numbered layers represent post-ReLU activations. It can be seen that before training, the CKA similarity between the two neural networks from layer 4 and onward is relatively low and the output being 0% similar while after training, the trained networks have learnt highly similar representation while their output being 98% similar.

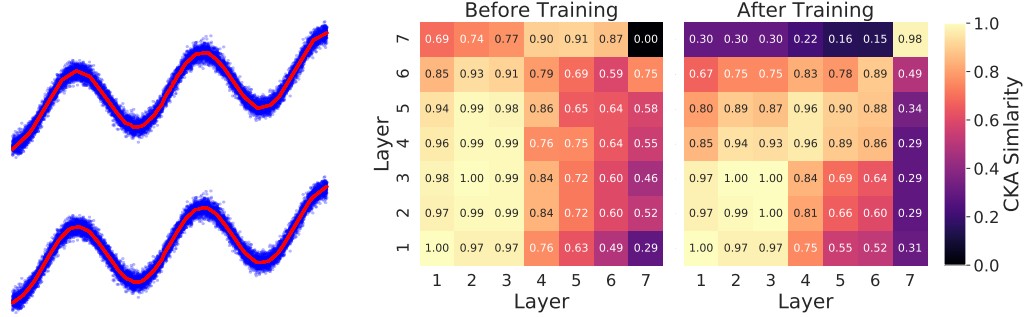

(a) Regression using two different neural networks

(b) CKA similarity heatmap between different layers of the two neural networks used for the regression experiment.

Figure 5: **Left:** Fitting a sine function using two different neural network architectures. The upper function was approximated using 64 neurons in each hidden layer while the lower function used 32 neurons in each hidden layer. **Right:** Represents the CKA similarity heatmap between different layers of both neural networks before and after training. The right diagonal (bottom left to top right) measures representation similarity of the corresponding layers of both neural networks. The trained networks have learnt similar representations while their output was 98% similar.

This example shows that neural networks can learn similar representation while trained on different batches. This observation is important because in MaxminDQN and EnsembleDQN training, each neural network is trained on a separate batch from the replay buffer but still learns similar representation similarity (see Figure 8).

# B  ALGORITHMS

For completeness, the regularizered MaxminDQN and EnsembleDQN algorithms are given below

---

**Algorithm 1:** Regularized MaxminDQN

---

The differences between the baseline MaxminDQN and regularized MaxminDQN are highlighted
Initialize $N$ Q-functions $\{Q^1, \ldots, Q^N\}$ parameterized by $\{\psi_1, \ldots, \psi_N\}$
Initialize empty replay buffer $D$
Observe initial state $s$
**while** *Agent is interacting with the Environment* **do**

  $Q^{min}(s, a) \leftarrow \min_{k \in \{1, \ldots, N\}} Q^k(s, a), \forall a \in \mathcal{A}$
  Choose action $a$ by $\epsilon$-greedy based on $Q^{min}$
  Take action $a$, observe $r$, $s'$
  Store transition $(s, a, r, s')$ in $D$
  Select a subset $S$ from $\{1, \ldots, N\}$  (e.g., randomly select one $i$ to update)
  **for** $i \in S$ **do**

    Sample random mini-batch of transitions $(s_D, a_D, r_D, s'_D)$ from $D$
    Get update target: $Y^M \leftarrow r_D + \gamma \max_{a' \in A} Q^{min}(s'_D, a')$
    Generate list of $L^2$ norms : $\boldsymbol{\ell} = \left[\|\psi_1\|^2, \ldots, \|\psi_N\|^2\right]$

    Update $Q^i$ by minimizing $\mathbb{E}_{s_D, a_D, r_D, s'_D} \left(Q^i_{\psi_i}(s_D, a_D) - Y^M\right)^2 - \lambda \mathcal{I}(\ell_i, \boldsymbol{\ell})$
  **end**
  $s \leftarrow s'$
**end**

---

**Algorithm 2:** Regularized EnsembleDQN

---

The differences between the baseline EnsembleDQN and regularized EnsembleDQN are highlighted
Initialize $N$ Q-functions $\{Q^1, \ldots, Q^N\}$ parameterized by $\{\psi_1, \ldots, \psi_N\}$
Initialize empty replay buffer $D$
Observe initial state $s$
**while** *Agent is interacting with the Environment* **do**

  $Q^{ens}(s, a) \leftarrow \frac{1}{N} \sum_{i=1}^{N} Q^i(s, a)$
  Choose action $a$ by $\epsilon$-greedy based on $Q^{ens}$
  Take action $a$, observe $r$, $s'$
  Store transition $(s, a, r, s')$ in $D$
  Select a subset $S$ from $\{1, \ldots, N\}$  (e.g., randomly select one $i$ to update)
  **for** $i \in S$ **do**

    Sample random mini-batch of transitions $(s_D, a_D, r_D, s'_D)$ from $D$
    Get update target: $Y^E \leftarrow r_D + \gamma \max_{a' \in A} Q^{ens}(s'_D, a')$
    Generate list of $L^2$ norms : $\boldsymbol{\ell} = \left[\|\psi_1\|^2, \ldots, \|\psi_N\|^2\right]$

    Update $Q^i$ by minimizing $\mathbb{E}_{s_D, a_D, r_D, s'_D} \left(Q^i_{\psi_i}(s_D, a_D) - Y^E\right)^2 - \lambda \mathcal{I}(\ell_i, \boldsymbol{\ell})$
  **end**
  $s \leftarrow s'$
**end**

---

# C  ALL TRAINING PLOTS

## C.1  TRAINING PLOTS FOR MAXMINDQN

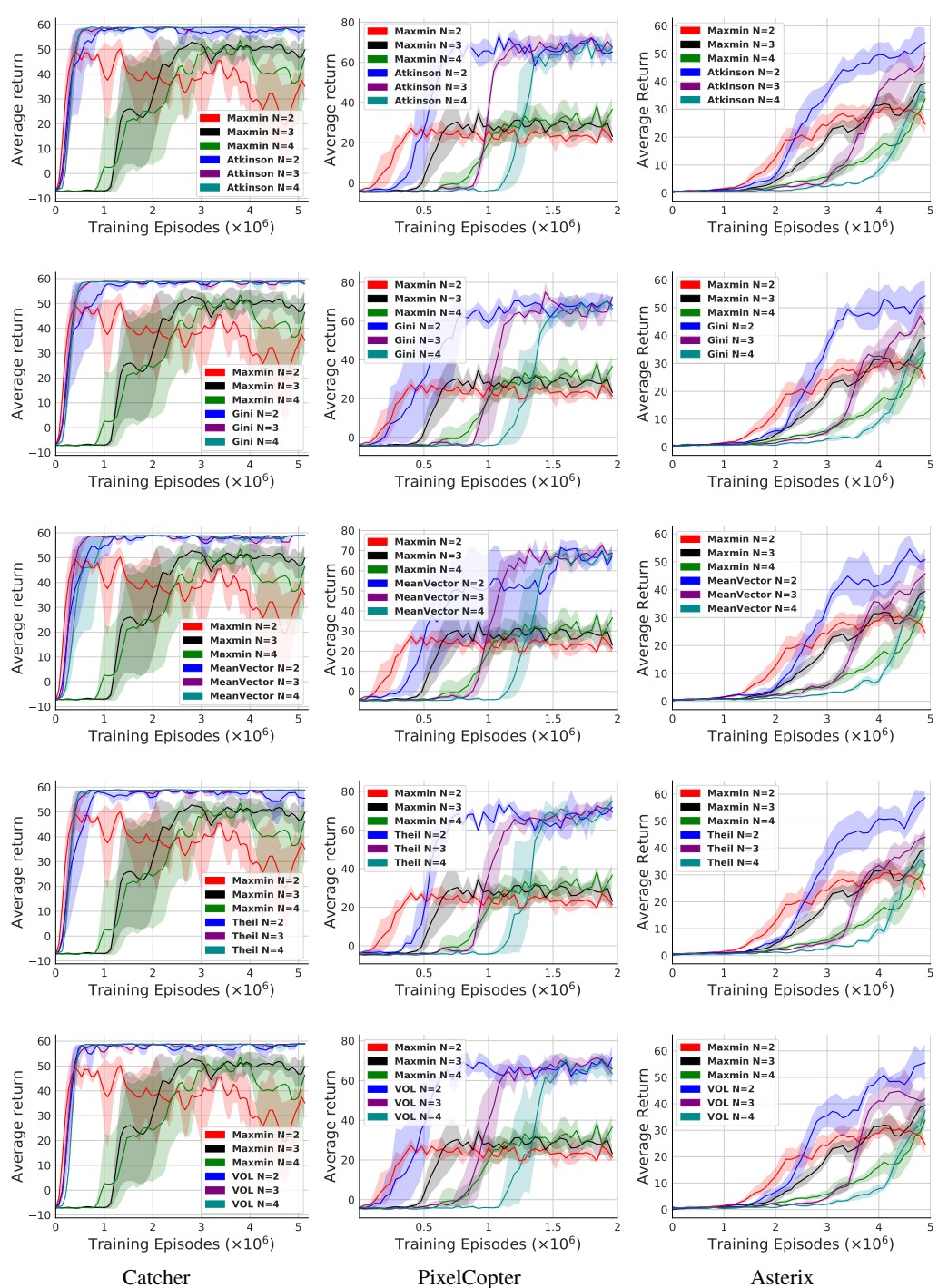

Figure 6: All MaxminDQN Results. Top to Bottom: Atkinson, Gini, MeanVector, Theil, Variance of Logarithms

## C.2 TRAINING PLOTS FOR ENSEMBLEDQN

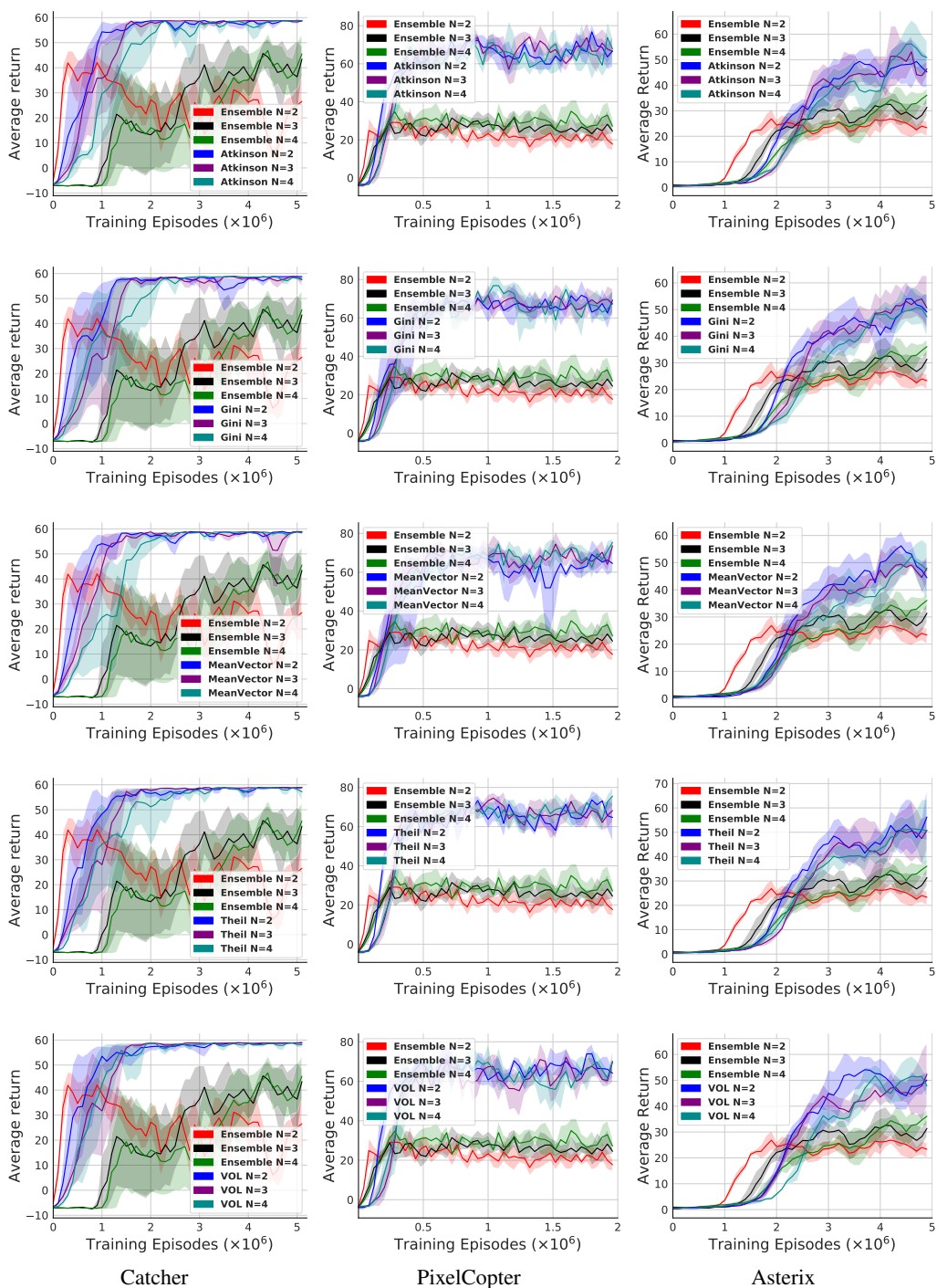

Figure 7: All EnsembleDQN Results. Top to Bottom: Atkinson, Gini, MeanVector, Theil, Variance of Logarithms

## C.3  HEATMAPS AND T-SNE VISUALIZATIONS

Figure 8 represents the CKA similarity heatmaps using a linear kernel of all the two neural network experiments after training averaged over all the five seeds. The point of interest is the right diagonal (bottom left to top right) that represents the representation similarity between corresponding layers. For the baseline experiments, the output layer has more than 96% similarity in almost all the scenarios while for the regularized versions have around 90% similarity in the output layer. This 10% difference provides enough variance in the Q-values to prevent the ensemble based Q-learning methods converging to the standard DQN.

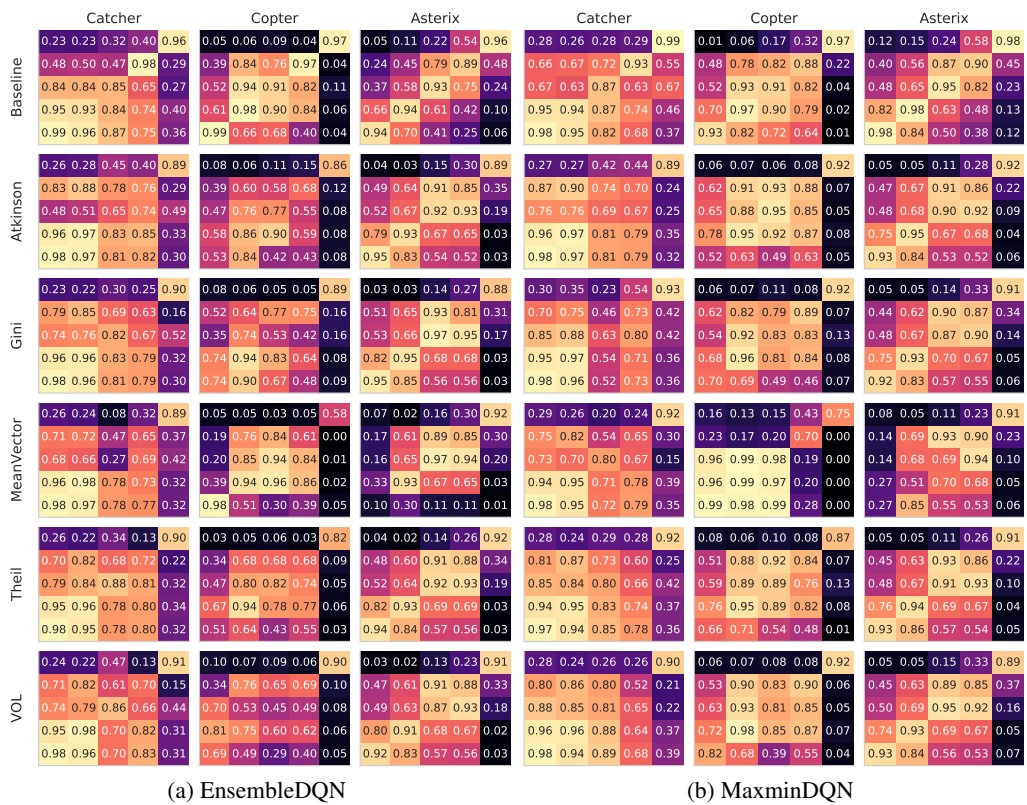

Figure 8: Heatmaps representing the CKA similarity of 2 neural network experiments.

Figure 9 represents the t-SNE visualizations of the baseline and regularized solutions trained with four neural networks on the PixelCopter environment. This visualization is consistent with the visualizations shown in Section 5.2 where the baseline activations are cluttered without any pattern while the Theil-MaxminDQN and Theil-EnsembleDQN activations have visible clusters.

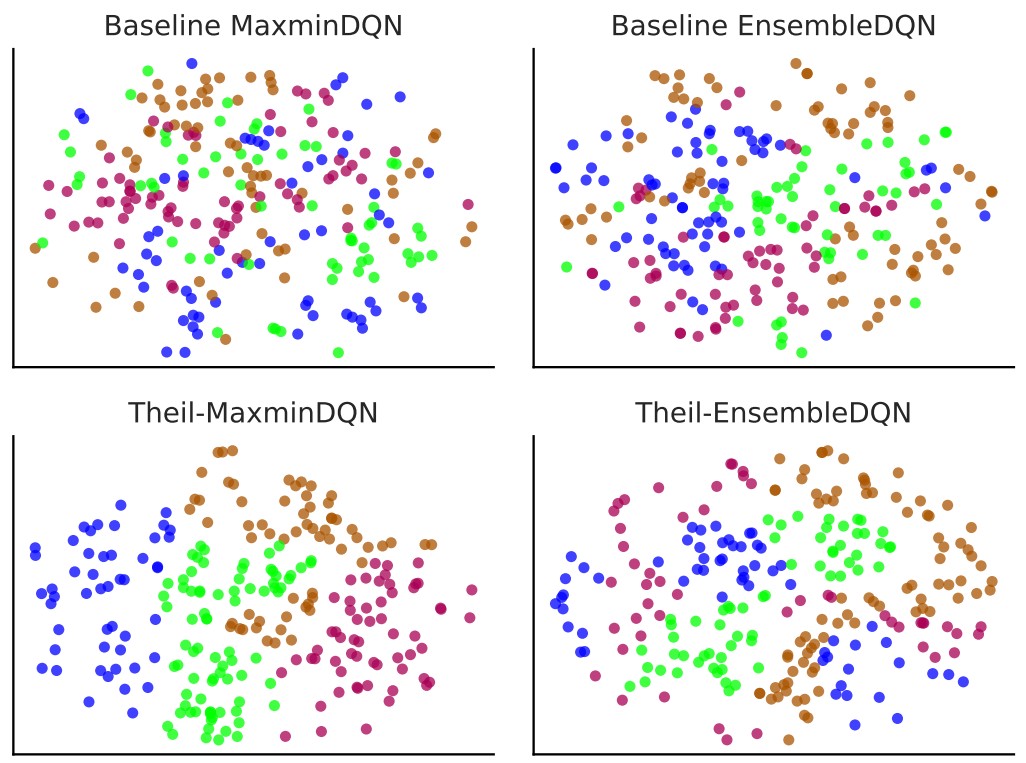

Figure 9: Clustering last layer activations from PixelCopter after processing them witht-SNE to map them in 2D

## C.4 z-Score Table for Four Neural Network Experiments

Table 3: Averaged $z$-scores for each regularization method with four neural networks

| Reg | Ensemble N=4 | | | Maxmin N=4 | | |
|---|---|---|---|---|---|---|
| | Catcher | Copter | Asterix | Catcher | Copter | Asterix |
| Baseline | -2.096 | -2.154 | -1.469 | -1.785 | -1.803 | -1.417 |
| Atkinson | **0.486** | 0.146 | 0.586 | 0.358 | 0.479 | **0.463** |
| Gini | 0.409 | 0.307 | 0.126 | 0.340 | 0.433 | 0.000 |
| MeanVector | 0.452 | **0.685** | 0.019 | 0.362 | 0.206 | 0.351 |
| Theil | 0.341 | 0.669 | **0.596** | **0.363** | **0.842** | 0.384 |
| VOL | 0.409 | 0.374 | 0.141 | **0.363** | -0.167 | 0.252 |

Table 4: $P$-values from Welch's $t$-test comparing the $z$-scores of regularization and baseline

| Reg | Ensemble N=4 | | | Maxmin N=4 | | |
|---|---|---|---|---|---|---|
| | Catcher | Copter | Asterix | Catcher | Copter | Asterix |
| Atkinson | 0.002 | 0.005 | 0.004 | 0.044 | 0.018 | 0.012 |
| Gini | 0.002 | 0.005 | 0.005 | 0.045 | 0.024 | 0.056 |
| MeanVector | 0.002 | 0.000 | 0.016 | 0.044 | 0.010 | 0.020 |
| Theil | 0.001 | 0.001 | 0.007 | 0.044 | 0.005 | 0.014 |
| VOL | 0.002 | 0.003 | 0.006 | 0.044 | 0.024 | 0.030 |

# D  IDENTICAL LAYERS EXPERIMENT (ALL TRAINING PLOTS)

## D.1  TRAINING PLOTS FOR MAXMINDQN

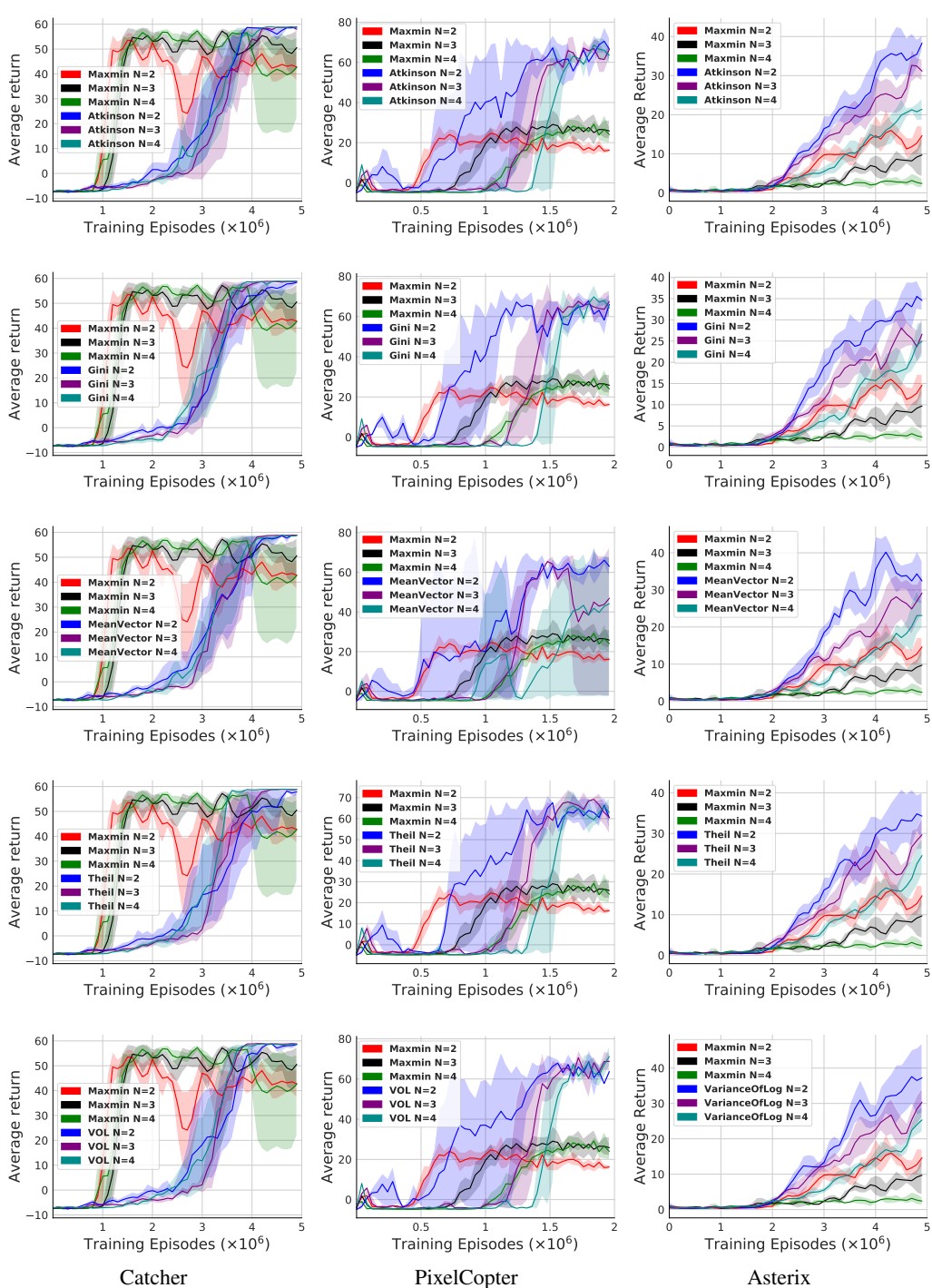

Figure 10: All MaxminDQN Results. Top to Bottom: Atkinson, Gini, MeanVector, Theil, Variance of Logarithms

## D.2 TRAINING PLOTS FOR ENSEMBLEDQN

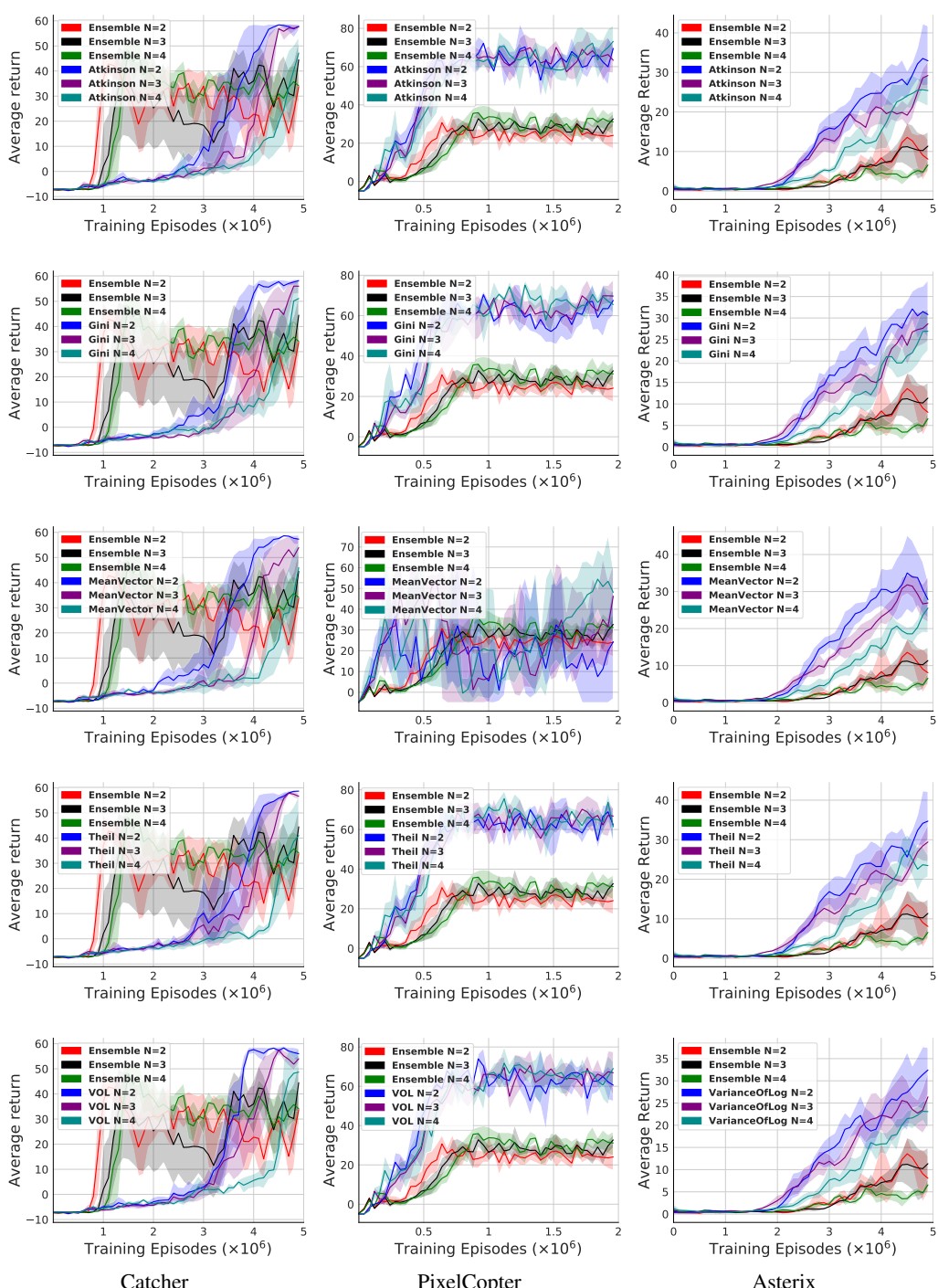

Figure 11: All EnsembleDQN Results. Top to Bottom: Atkinson, Gini, MeanVector, Theil, Variance of Logarithms

# E    IMPLEMENTATION DETAILS AND HYPERPARAMETERS

For our implementation of MaxminDQN and EnsembleDQN we used the code provided by the MaxminDQN authors that has implementations of different DQN based methods (github.com/qlan3/Explorer). For the baseline experiments, we used most of the hyperparameter settings provided in the configuration files by the authors except learning rates which we limited to $[1e-3, 1e-4, 3e-5]$ and limited the number of ensembles to four. The complete list of hyperparameters for each environment is shown in Table 5. The values in bold represent the values used for the reported results.

Table 5: List of hyperparameters used for the experiments.

| Hyperparameter | Catcher | PixelCopter | Asterix |
|---|---|---|---|
| Learning rate | $[1e-3, \mathbf{1e-4}, 3e-5]$ | $[\mathbf{1e-3}, 1e-4, 3e-5]$ | $[1e-3, 1e-4, \mathbf{3e-5}]$ |
| Batch size | $[\mathbf{32}, 64]$ | $[32, 128, \mathbf{1024}]$ | $[32, \mathbf{64}]$ |
| Buffer Size | $[1e4, \mathbf{1e7}]$ | $[1e4, \mathbf{1e6}]$ | $[1e5, \mathbf{2e5}]$ |
| Exploration Steps | $\mathbf{1e3}$ | $[1e3, \mathbf{2e3}]$ | $\mathbf{2e5}$ |
| Hidden Layer Size | $[\mathbf{64, 64}]$ | $[\mathbf{64, 64}]$ | $[\mathbf{64, 64}]$ |
| Gradient Clip | $\mathbf{5}$ | $\mathbf{5}$ | $\mathbf{-1}$ |
| Discount Factor $\gamma$ | $\mathbf{0.99}$ | $\mathbf{0.99}$ | $\mathbf{0.99}$ |
| Regularization Weight | $[1e-5, \mathbf{1e-6}]$ | $[1e-6, 1e-7, \mathbf{1e-8}]$ | $[1e-5, \mathbf{1e-6}]$ |

For the identical layer experiment, no hyperparameter tuning was performed and we reused the hyperparameters from the main results. In terms of number of experiments, we ran 190 experiments: (5 regularizers $\times$ 5 seeds $\times$ 3 ensemble settings $\times$ 2 algorithms) + 40 baseline experiments for each environment totaling 570 runs for all environments after hyperparameter tuning. The same number of experiments were performed for the identical layer experiment which sums up to 1140 runs where each run took 11 hours of compute time on average.

# F    PLOTTING THE GINI INEQUALITY

We measured the $L^2$ norm inequality of the baseline MaxminDQN and EnsembleDQN along with their regularized versions. We trained baseline MaxminDQN and EnsembleDQN with two neural networks along with their Gini index versions with regularization weight of $10^{-8}$ on the PixelCopter environment on a fixed seed . Figure 12 represents the $L^2$ norm inequality of the experiments along their average return during training. Notably, despite each neural network being trained on a different batch, the $L^2$ norm of the baseline MaxminDQN and EnsembleDQN are quite similar while the $L^2$ norm of the regularized MaxminDQN and EnsembleDQN have high inequality.

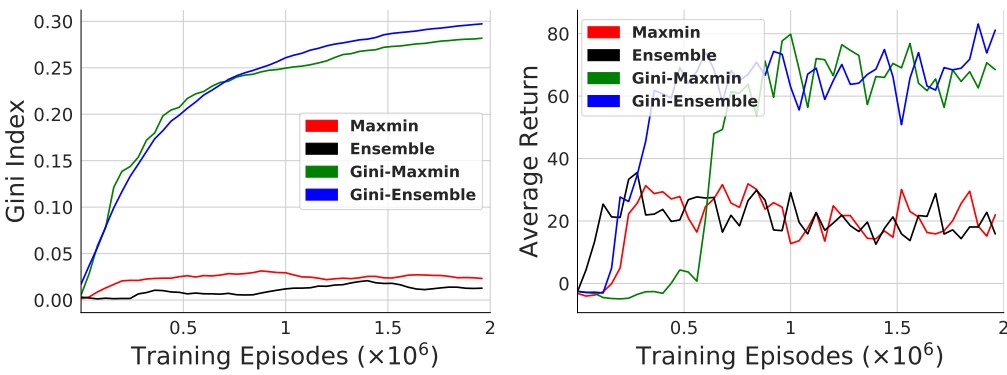

Figure 12: **Left**: Plot representing the $L^2$ norm inequality between the two neural networks using Gini index trained on PixelCopter environment. **Right**: Plot representing the average return during training.

