# OpenReview forum: "Preventing Value Function Collapse in Ensemble  Q-Learning by Maximizing Representation Diversity"
_ICLR.cc/2021/Conference — Reject_

### Official Review · AnonReviewer4 · 2020-10-25
**Review for "Preventing Value Function Collapse in Ensemble Q-Learning by Maximizing Representation Diversity"**

**Rating:** 4
**Confidence:** 1

**Review:**

Summary:
The paper presents different regularizers that promote representation diversity in order to prevent value function collapse and lead to better performance. This is mainly based on a conjecture and is verified with an experimental study.

General comments:
The paper is well written and presents the different concepts quite well. I think the experimental section is well organised. However the number of environments is quite small. I do think that the contribution would be much stronger with a more exhaustive study. At the moment, it is quite hard to have a definite answer on how much the regularizers proposed are useful. One thing that could strengthen the experimental section is to provide the number of weights used for each methods. I assume that the ensemble networks used N times more weights that the simple DQN. If so, would it be possible to compare with a network N times bigger for DQN?. It is mainly to compare for a fixed computational budget how much the proposed methods are better.

Finally, overestimation may not be all bad in particular for exploration. It could be a way to allow for some optimism in the face of uncertainty. It would be nice to show if this intuition is wrong and provide more evidence why overestimation is wrong and lead to bad outcomes. This could be done with a broader experimental study.

Rating: I am ready to change my rating if some or all of my comments are answered. This review is an educated guess and I am not at all an expert in the field of overestimation in statistics, economy or reinforcement learning.

---

### Official Review · AnonReviewer1 · 2020-10-27

**Rating:** 5
**Confidence:** 4

**Review:**

This paper proposes methods to induce diversity in the networks of ensemble-based Q-Learning methods. This is achieved my maximizing a variety of measures of inequality based on the L2 parameter norms of individual networks in an ensemble. This is motivated by the benefit of having diversity in the learned features, which itself is motivated by observations on the CKA of some ensembleDQN networks.

Strengths:
- The high-level motivation of this work is sound. Diversity within an ensemble is undeniably desirable
- The proposed methods do improve performance on interesting benchmarks

Possible improvements/Weaknesses:
- There are missing steps in the chain from motivation to method, but these steps can easily be done and verified by measuring the appropriate quantities systematically.
- It's not clear exactly what the method is doing, it would be good to actually measure correlations between feature similarities and performance (which the authors claim exists without measuring)
- It's not clear if the method is truly benefiting from feature diversity, or if the regularization induces some totally different effect, simple baselines and using a variety of feature similarity metrics could address this

I gave this paper a score of 5; while the method is somewhat interesting, the authors need to **quantitatively** show that the regularizations they propose have the effects they claim they have. Otherwise, this is just a "my number is bigger than your number"-paper, which isn't very valuable for our scientific understanding of deep RL.


Detailed comments:
- "The first deep RL algorithm, DQN", that's not very historically accurate. Martin Riedmiller published Neural Fitted Q Iteration in 2005, Gerald Tesauro published TD-Gammon in 1995, and I don't even know if they're really the "first". Even if we're stricter about the notion of "deep", in 2010 Lange & Riedmiller successfully trained deep autoencoders to pretrain features used for NFQ, in 2012 Hess & al used RBMs and MLPs to model policies. Perhaps "landmark" or "remarkable" would be more accurate.
- The authors emit the hypothesis that "representation similarity between neural networks in an ensemble-based Q-learning technique correlates to poor performance. To empirically verify our hypothesis...", but that hypothesis is never verified, showing qualitative plots of two networks with high CKA is very different than making a clear measurement of correlation. For example, a figure with CKA on the x axis and performance on the y axis for repeated runs and a clear correlation line and r^2 value would be a quantitative statement testing the hypothesis.
- "that random initialization of neural networks enforces diversity is a misconception". It's known that different similarity metrics like CCA or maximum matching tend to find that two networks trained with different initializations _do_ converge to different features, while CKA finds they do _not_. It might be valuable to test these metrics in the RL setting as well, I suspect the conclusions/motivation of this paper might differ somewhat.
- Figure 3 is quite misleading, in the bottom row the individual networks are trained to have different norms, thus it is extremely likely that the activations of the last layer are also going to have different norms. This is the first thing the t-SNE is going to pick on, and in that sense Figure 3 most likely isn't showing that different features have been learned, and it's also likely that if the networks learned similar features with different scales t-SNE wouldn't be able to show it.
- The argument the authors make about feature diversity is based on CKA, but the methods are based on L2 norms of parameter vectors.
  1. The link between the two is not made (and not obvious to me)
  2. Deep ReLU networks are known to be invariant under several reparameterizations, including some rescalings (see Dinh et al, https://arxiv.org/abs/1703.04933). It's not clear to me that two networks that have different L2 norms have significantly different features. It's in fact quite easy to take a trained network and then finetune it only with L2 or L1 without significant loss in performance (presumably the same features are thus kept, just scaled; this is what pruning, quantization, and DNN compression are based on). I thus don't see how diversity in features follows from diversity in L2 norms.
  3. A crucial missing baseline seems to be to train an ensemble of networks with a varying L2 regularization loss on each network of the ensemble. Another interesting baseline would be to vary the learning rate of each network, since having smaller or larger parameters may in the end only affect the effective learning rate of a particular network.
- The results of 5.1 and Fig 2 are interesting, but
  1. it would be interesting to see the average performance of diversity-regularized methods, not just the best ones
  2. it seems like a downside that no single diversity method is systematically the best, but perhaps it reflects something about the environments used, and could be interesting to investigate.
- The conclusion claims that "high representation similarity between the Q-functions in ensemble[s] [..] leads to a decline in learning performance". The paper only seems to have 4 data points to that effect, Figure 1. This is not a very strong claim.
- The conclusion claims that the proposed method can "maximize the diversity in the representation space", but this is not measured, only hypothesized. The claims that this paper really makes are:
  - CKA can measure diversity, and ensembles with high CKA perform poorly (this is suggested by Figure 1 and 8, but not measured precisely), thus low diversity is bad
  - Having variety in L2 norms in an EnsembleDQN setup is good
 One big missing step is to verify that diversity in L2 norms induces diversity as measured by CKA. Another missing step as I mention earlier is that to show that high diversity (as measured by CKA) induces good performance.
- Appendix F should be much larger, and contain a plots of all 5 diversity measures, as well as, ideally, individual or group statistics of the L2 norms these regularizations are acting upon.

---

### Official Review · AnonReviewer3 · 2020-10-29
**Borderline paper, improved discussion / investigation of effects could help significantly**

**Rating:** 5
**Confidence:** 4

**Review:**

This paper proposes to improve ensemble-based methods such as Maxmin
and Ensemble Q-learning by regularizing the representations for the
ensemble members to increase diversity. The motivating intuition is
that on their own, these methods can still collapse and once they do
the benefits of the ensemble are lost. The authors use a variety of
measures of inequality to provide regularization (which increase that
inequality of representation among ensemble members).

Although the hypothesis seems plausible, I don't think the results
sufficiently support it.
In particular, much of the motivation focuses on over-estimation, but
this was never evaluated explicitly. I think the experimental results
could be strengthened in a number of ways: more domains, single set of
hyperparameters, further empirical analysis of the effects of the
different inequality measures used for regularization.

Pros:

Interesting idea and seems effective based on experimental results.
Highly relevant, as ensembles are one of the most commonly used and
best methods for a number of use cases in and outside of RL.

Cons:

Per-domain hyperparameter tuning
Small set of domains
Limited understanding of  why some regularization methods work better
than others.

Comments/Questions:

Section 4.1-4.2
Figure 1 did not convince me about the similarity measure the way I
think the authors wanted it to. Just looking at the Heatmaps, I really
struggled to see why A and C should be better than B and D. The only
view on these I could come to was that if you only look at the
diagonal components (or center block) of the heatmaps then the
connection works. But, why are these the similarities that matter?
Overall this section (4.2) does not lay the groundwork of motivating
the correlation between performance and representation similarity the
way it was intended to. It seems like this could have been validated
much more directly, by controlling the representations used first, and
then measuring performance afterwards.

Section 5.1
Hyperparameters appear to have been tuned per-domain, which makes me
sceptical about the results as a whole. A more typical approach would
be to use a single set of hyperparameters over the set of domains
considered. One reason for this is that doing so penalizes methods
that are sensitive to hyperparameter tuning, which should be
considered as part of the algorithm itself.

Regularization does seem effective, but these are very limited
results. Why not increase to the full set of MinAtar games?

Section 5.2
What do the colors represent?
As this section attempts to give us some idea of the impact of
regularization (beyond simply performance) I would have really liked
to get more intuition for the ways the regularizers are affecting the
representation more directly. The t-snes are nice to show something is
happening, but don't give us any insight into how or why. How should
we understand the differences in the different methods used? Why
should one be preferable over another?


Typos:

Page 2, "The implicit policy \pi can extracted" (can be extracted)
Page 7, "we calculated the z-score for each regularization method and
the baseline by treating results all the results" (remove the first
'results')

---

### Official Review · AnonReviewer2 · 2020-10-29

**Rating:** 6
**Confidence:** 3

**Review:**

Q-learning is known to have overestimation bias. Approaches like EnsembleDQN and MaxminDQN try to use different estimates from ensembles of learners to reduce the bias. The authors study a specific observation and try to tackle it by regularization technique to maximise the diversity of representation space. Five different regularization functions are evaluated in the paper. And experiments show that the proposed regularization helps on the diversity and outperform MaxminDQN and EnsembleDQN. Note that the reviewer is not very familiar with methods to introduce diversity in representation, but based on educated guess, the proposed method look interesting.

Pros:
+ The reviewer really likes how the paper goes from Section 4.2, illustrating the misconception of using different random initialization of neural networks could enforce diversity, then going to 4.3 to list the potential regularization metric which could be borrowed from economic theory and consensus optimization, including Atkinson Index, Gini coefficient, etc.
+ Results in Section 5.1 and Figure 2 demonstrate the excellent performance of the proposed method;
+ The t-SNE visualizations in Figure 3 are very clearly demonstrating the effect the regularization has;

Cons:
- From results in Figure 2, the best perform regularization methods keep changing, e.g., for (a) Catcher, they are Gini-Maxmin and VOL-Ensemble and for (b) PixelCopter they are VOL-Maxmin and Theil-Ensemble. It would be great if the authors could keep the algorithm fixed and then show the performance across different games.

---

> ### Author Response · Authors · 2020-11-13
> **Detailed Training Plots**
>
> >It would be great if the authors could keep the algorithm fixed and then show the performance across different games.
>
>
> Thank you for comments. For the main results section, we chose to show the best results for each algorithm and neural network settings. This was done to avoid the clutter and confusion. The detailed training plots as your recommendation are already shown in Appendix C, Figure 6 and 7.

---

> > ### Comment · AnonReviewer2 · 2020-11-24
> > **Reply**
> >
> > Sorry that I missed the plots Figure 6 & 7 in the Appendix. Thanks for pointing that out.

---

### Decision · Program_Chairs · 2021-01-07
**Final Decision**

**Decision:**

Reject

**Comment:**

The paper tackles the Q-value overestimation problem by proposing a regularization technique to maximize diversity in representation space, preventing ensemble "collapse", in order to improve the efficacy of techniques such as Maxmin and Ensemble Q-learning. Reviewers praised the originality of the method and the interesting connections drawn to economic theory, and seemed to agree that the method is somewhat effective. On the other hand, R3 pointed out that hyperparameters are tuned per-domain, the number of domains considered is small (echoed by R4), and criticized the paper for failing to truly validate its central hypothesis experimentally (echoed by R1).  R4 raised the issue of unfair comparisons in between ensemble and non-ensemble methods, while R1 raised a multitude of criticisms ranging from ahistorical attributions to confusing figures, which I will not exhaustively repeat here. Based on the reviews, and the fact that the majority of reviewers' concerns remain entirely unaddressed (the authors only responded to R2), this manuscript is not a candidate for acceptance at this time.